# Measuring the Performance of Private Pension Companies in Türkiye by Gray Relational Analysis Method

**Muharrem Umut** 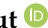

Department of Insurance, University of Ankara Hacı Bayram Veli, Ankara 06000, Türkiye;
muharrem.umut@hbv.edu.tr

**Abstract:** The private pension is a system designed to maintain an income level during passive periods by utilizing the income earned during active working years. It complements the mandatory retirement systems of the public sector and is based on a voluntary participation structure. Additionally, it serves as an investment and savings tool with the ability to provide long-term funds. The legislation for the private pension system was enacted in Türkiye in 2001, and it was implemented in 2003. In addition, a government contribution program was initiated to promote the system in 2013. An automatic enrollment system was introduced in 2017. The effectiveness and performance of individual pension companies play significant roles in the system. This study aims to measure the performance of individual pension companies operating in Türkiye using the gray relational analysis method, which is an effective measurement method, for the years 2016–2022. Subsequently, based on the measurement results, recommendations will be provided.

**Keywords:** private pension system; pension companies; public authority; government contribution; gray relational analysis method

## 1. Introduction

The need for individuals to have an additional source of income emerged in response to low savings and investment levels, improvements and increases in health conditions, a decrease in mortality rates, and an improvement in living standards in our country. In 2001, the private pension system (PPS), which was recognized as a significant reform, underwent legislative regulation and was implemented in 2003. Prior to that, Türkiye experienced the 1999 earthquake, and due to economic problems, individuals had a low inclination toward saving, leading to the search for alternative options. Consequently, the PPS emerged as a system aimed at enabling individuals to maintain the level of prosperity they experienced during their active working years in the retirement period, also known as the passive period, serving as a savings tool. Fundamentally, the PPS complements the mandatory retirement systems of the public sector and operates on a voluntary basis. Furthermore, the PPS serves as an investment instrument during the active period (Can and Eyidiker 2019). It was first introduced in our country as part of the reform in the public social security system in 2001 and acquired a legal framework with the Private Pension Savings and Investment System Law.

Retirement systems have a three-tier structure based on common practices worldwide. These tiers consist of mandatory public retirement systems (such as social security institutions and private social security funds), employer-based retirement programs (such as Oyak, İlksan, and Polsan, which are organizations subject to specific laws, foundations, commercial companies, associations, and funds governed by Law No. 4632, employer group pension contracts, and automatic enrollment systems to private pension plans), and lastly, private pension systems based on voluntary participation, such as private pension systems (İçöz and Özdemir 2018). Retirement services can be provided separately from mandatory public systems, allowing for separate determination of contribution rates, deduction rates, and retirement conditions. In other words, by making the system flexible,

retirement plans can be created by prioritizing individuals' preferences and choices and providing specific benefits and conditions. The structure of the PPS has been designed in this way. Therefore, individuals have the freedom to enter alternative systems by considering their benefits.

The need for private pension systems primarily arises from factors such as the development of healthcare services, a decrease in mortality rates, an aging population, an increase in living standards, and population growth, with the aim of maintaining and enhancing individuals' welfare by preserving their income levels. Additionally, PPS serves as an important alternative for closing the actuarial deficit or reducing the burden on the public sector, which refers to the inability of the income earned by active workers to cover the expenses of inactive individuals (retirees). Currently, mandatory public retirement systems are experiencing actuarial deficits and exert significant pressure on employees (Billig and Ménard 2013). The actuarial deficit is attributed to optimistic return assumptions, early retirement arrangements, and increased life expectancy.

Furthermore, the low savings level is also a significant factor. Increasing the savings level through PPS contributes partially to closing the actuarial deficit. In summary, while PPS is important for closing the mentioned deficit in the public sector, it also serves as an investment and savings tool for individuals, assuming a significant role in countries. However, our country experiences a continuous increase in the elderly population, with an estimated population of approximately 98 million by the year 2050 (Turkish Statistical Institute 2023). This is illustrated in Figure 1 below.

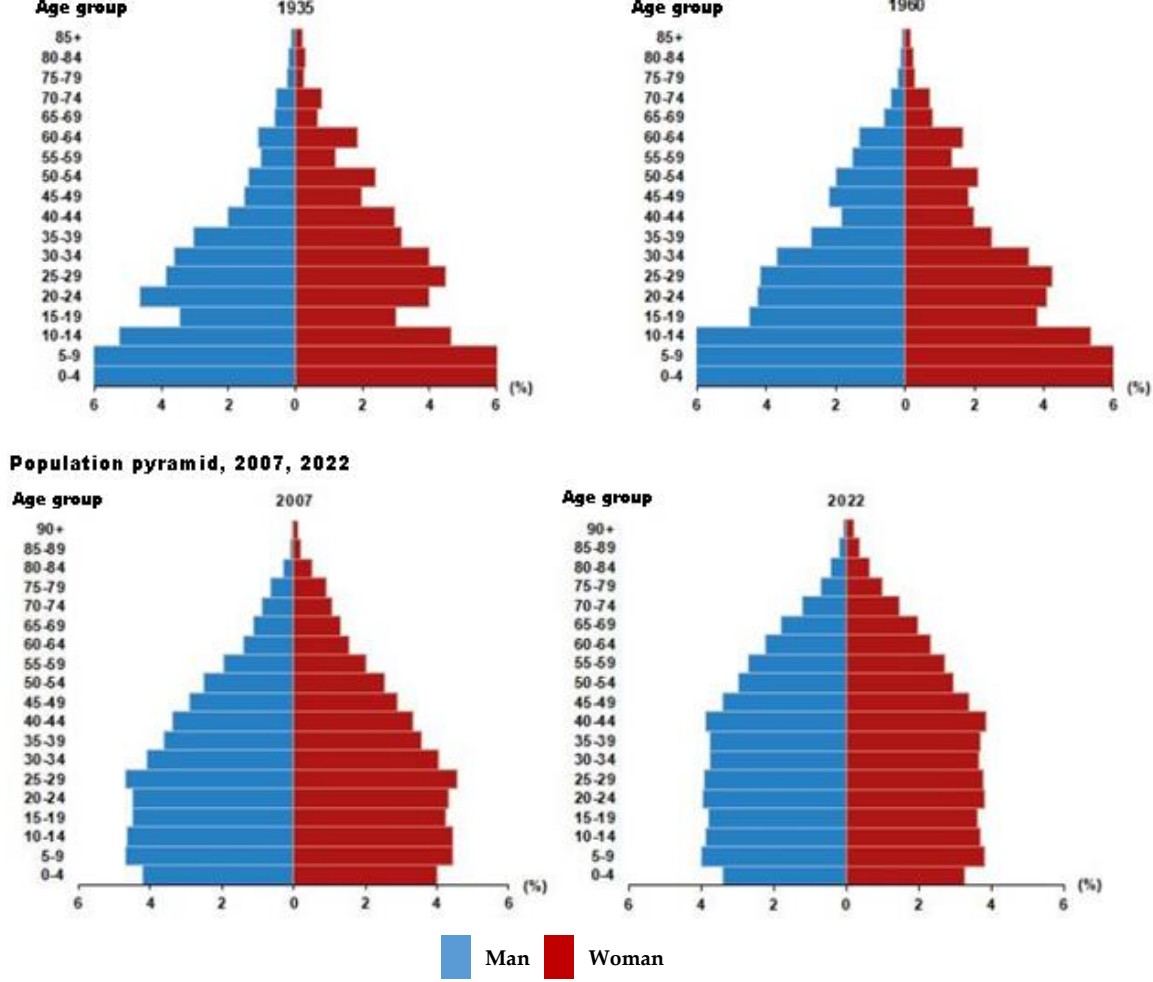

**Figure 1.** Population Development of Türkiye Over Time. Source: https://data.tuik.gov.tr/ (accessed on 19 March 2023).

Taking into account the mentioned reasons, the PPS was implemented in our country as a significant reform with the Private Pension Savings and Investment System Law in 2001, and the system commenced its operations in 2003. Subsequently, revisions and innovations were made to the system for various reasons. First, to increase interest in the PPS and provide tax incentives, the state contribution program was introduced in 2013. Additionally, due to the failure to achieve the desired goals, low savings levels, and an increase in the elderly population, the automatic enrollment system was introduced in 2017. Thus, the reform process or stage of the PPS can be divided into three phases: the implementation and development of the PPS in 2001, the period of state contribution in 2013, and the introduction of the automatic enrollment system in 2017 (Umut 2020). The distinctive feature of these reforms is that the system experienced relative growth with state contributions and expanded its coverage with the automatic enrollment system.

As of the end of 2022, there were 15 pension companies, 380 funds, and 71,462 individual pension agents in our country. The total fund size is approximately TRY 419 billion, and the number of PPS participants is 8 million. The number of automatic participants is 6.7 million, with a fund size of around TRY 35 billion. The average monthly contribution amounts are TRY 448 in the voluntary PPS system and TRY 200 in the automatic enrollment system. The proportion of participation-based funds is approximately 26% (Pension Monitoring Center 2023).

In the study, first, the importance of PPS, its basic features, and its function will be discussed, along with the state contribution. Subsequently, the automatic enrollment system will be addressed, and the new regulations in PPS will be mentioned. Then, the study will proceed to the application section, where the performance of individual pension companies will be measured using the gray relational analysis method. Finally, the application results will be evaluated, conclusions will be drawn, and insights will be provided for future studies.

In the conducted literature review, we observed that the data envelopment analysis method is commonly used for performance measurement. However, due to reasons such as the simplicity of the data envelopment analysis method, obtaining significantly different results with minor deviations, and the method yielding unreliable results in the face of new input–output entries included in the set, the gray relational analysis method, known to be more effective and innovative, has been employed. Indeed, in various sectors such as finance, aviation, transportation, and supply chain, this analysis method is utilized.

In the insurance sector, an integral part of the finance industry in Türkiye, the performances of pension companies are crucial factors in participants' preferences for these companies. The higher the financial efficiency and performance of a company, the more it will be preferred by individuals. In the measurement of the performance of individual pension companies, six main indicators have been utilized: equity, total assets, number of participants, total participant fund amount, pension technical revenue, and pension technical profit/loss. Although there are numerous financial ratios and indicators used in insurance companies, within the scope of this study, these indicators are considered sufficient for the measurement of pension companies.

As a result of the gray relational analysis conducted, there is a decline in companies' performance ratios over the years, and while the number of participants in certain insurance companies decreases on a company basis, specific companies consistently rank high in performance efficiency. It is evident that the company exhibiting the best financial performance ensures efficient management in terms of equity, total assets, number of participants, participant fund amount, pension technical revenue, and pension technical profit/loss. The obtained results align with existing studies in the literature. The variables used in the study were analyzed based on the data envelopment analysis method for the same years and yielded similar conclusions.

## 2. Private Pension System (PPS): Basic Features and Functions

The system is fundamentally based on voluntary participation and complements the mandatory pension system in the public sector. PPS provides individuals with the opportunity to generate a second income to maintain their living standards during their active working years. It is a funded system based on personal accounts, and the funds remain within the country. It has an important function in instilling saving habits in people, thereby also providing the government with long-term funding opportunities. In terms of organizational structure, PPS involves various functions such as pension companies, intermediaries, custodian institutions, specialized portfolio management companies, and supervisory and regulatory bodies.

However, the primary function of PPS is to direct the contributions saved by individuals during their active working years into investments, ensuring the continuity of the welfare level during retirement (Apak and Taşçıyan 2010).

There are two basic requirements to retire under the PPS. The first is to reach the age of 56, and the second is to remain in the system for at least 10 years. Individuals who retire can choose to receive a lump-sum payment or opt for programmed repayment through an annual income insurance. However, participants can withdraw from the system with certain deduction rates before meeting these conditions. Anyone with legal capacity can directly participate in the system, while those without legal capacity can participate through a parent/guardian by purchasing a plan from a pension company. In the PPS, there are three types of deductions: entry fee, fund management expenses, and administrative expenses. However, with the amendment made in the PPS legislation in 2016, a favorable approach was adopted for individuals in the automatic enrollment system, limiting the deduction to only fund management expenses (Regulation on Individual Pension System 2012).

Participants have the flexibility to change the amount of contributions they make as long as it does not fall below the predetermined minimum level. They can also switch to another pension company, redirect their savings to another company, consolidate their plans, and pause payments when necessary. They can change the distribution ratio of funds up to 12 times a year and change their plans up to 4 times. There is a right of withdrawal within 60 days in the individual pension contract, and when this right is exercised, the entire entry fee and administrative expenses are refunded. These two deduction items are not made for the sixth year of the contract and beyond (Regulation on Individual Pension System 2012).

After these deductions, contributions are invested in investment funds. The decision of which funds to invest in is entirely up to the participants themselves. Moreover, these funds are managed by portfolio management companies specializing in the field. The assets transferred to the funds are held by Takasbank, the custodian institution. Individuals have the opportunity to track their funds and returns through Takasbank (Law on Private Pension Savings and Investment System 2001).

Additionally, institutions in the system provide significant services depending on their areas of activity. For example, the Insurance and Private Pension Regulation and Supervision Agency (IPPRSA) handles supervision and regulation, the Capital Markets Board (CMB) is responsible for the establishment and administrative procedures of funds, the Ministry of Treasury and Finance handles state contribution payments, the Pension Monitoring Center (PMC) manages the information systems and supervision functions for collecting and managing data, and Takasbank and portfolio management companies play important roles in the system.

In Table 1 below, basic descriptive data within the scope of PPS are presented.

As of the end of 2022, it can be observed that the fund amount of the private pension system (PPS) has reached TRY 351.4 billion, and the amount of government contributions has reached TRY 48.6 billion. The number of participants is approximately 7.8 million individuals. The fund amount in the automatic enrollment system has reached TRY 15.7 billion, and the number of participant certificates is approximately 9.5 million. The

total fund amount of the PPS and automatic enrollment system is approximately TRY 367 billion (Pension Monitoring Center, www.egm.org.tr, accessed on 19 March 2023).

**Table 1.** PPS Basic Indicators (2022).

| Number of PPS Participants | 7,801,313 |
|---|---|
| PPS Fund Size | 347.7 Billion TRY |
| PPS State Contribution Fund | 50.1 Billion TRY |
| Number of Automatic Enrollment System Certificates | 9,488,458 |
| Automatic Enrollment System Fund | 31.1 Billion TRY |
| Number of Active Private Pension Agents | 71,462 |
| Number of Pension Companies | 15 |

Source: Pension Monitoring Center (www.egm.org.tr), accessed on 19 March 2023.

As seen from the Figure 2, the PPS reached saturation between the years 2017–2021, and we observed that the number of participants was stuck in this range because of the more advantageous automatic enrollment system put into effect in 2017, economic fluctuations, and the effect of the pandemic. In other words, although there were fluctuations, there was no significant change in the number of participants of the PPS. However, the system gained momentum again as a result of two important developments: the opportunity to be included in the system for participants under the age of 18 (who do not have the capacity to act) in 2021 and the increase in the state contribution rate from 25% to 30% at the beginning of 2022. Although there is a reflection of the increase in the number of participants in the fund amount, the return in capital instruments, which gained value especially in an inflationary environment, is the main reason for the high increase in 2022.

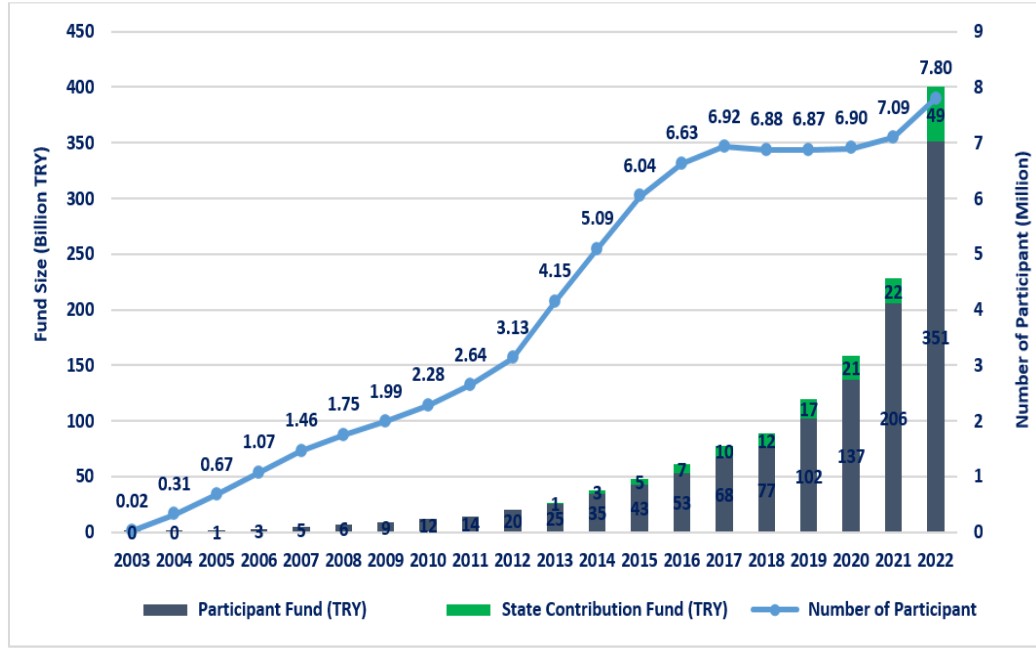

**Figure 2.** PPS Participant Count and Fund Size Over the Years (2003–2022). Source: Pension Monitoring Center (www.egm.org.tr), accessed on 19 March 2023.

## 3. Government Contribution in the Private Pension System (PPS)

In 2012, a significant amendment was made to the PPS law with the aim of increasing the desired fund level and participant count, maximizing the utilization of tax incentives and accelerating the momentum of the system. As a result, a government contribution

was introduced to the system. This contribution is provided by the government, covering 25% of the participant's contributions when they join the system. In other words, 25% of the contributions are transferred directly from the government to the individual accounts of the participants. For example, if a participant contributes TRY 1000 to the system, TRY 250 will appear in their account as a government contribution. This amendment was intended to enhance interest in the PPS and ensure a revival similar to its initial introduction. However, due to the failure to reach the desired participation count and fund size, the government contribution rate was increased to 30% through an amendment to the Law on Private Pension Savings and Investment System (No. 4632) on 22 January 2022. The 30% government contribution is separately tracked and limited to specific investment instruments and funds for evaluation. Therefore, participants do not have the ability to intervene in the 30% government contribution. However, certain limitations (entitlements) are imposed in order to qualify for the government contribution.

The entitlement percentages for participants to be eligible for the government contribution are presented over the years in the Table 2. For instance, if a participant remains in the system for 3–6 years and then withdraws, they are entitled to 15% of the government contribution. If they stay in the system for 6–10 years, the entitlement rate is 35%, and for 10 years or more, it is 60%. Finally, if they fulfill the necessary requirements and retire, they are entitled to 100% of the government contribution.

**Table 2.** Entitlement Rates After Government Contribution.

| Time Spent in the System after January 2013 | Entitlement Percentage |
| --- | --- |
| 3–6 Years | 15% |
| 6–10 Years | 35% |
| 10 Yıl+ | 60% |
| In Cases of Retirement or Death or Disability | 100% |

Source: Regulation On State Contribution In The Individual Pension System, 2012 (www.egm.org.tr), accessed on 19 March 2023.

Table 3 presents statistical data on participants, companies, government contributions, and the amount directed to investments in the private pension system (PPS) after the government contribution, based on current data. The table provides a breakdown of these figures to illustrate the distribution among participants, companies, and the amount allocated to investments.

**Table 3.** PPS Funds Amounts (2022).

| | |
| --- | --- |
| **Fund Amount of Participants** | **TRY 347,685,208,737** |
| Interest Fund Amount | TRY 268,557,206,621 |
| Interest-Free Fund Amount | TRY 78,128,002,116 |
| **State Contribution Fund Amount** | **TRY 50,072,987,034** |
| Interest State Contribution Fund Amount | TRY 44,004,685,532 |
| Interest-Free State Contribution Fund Amount | TRY 6,068,301,502 |
| **Total of Companies' Participants** | **7,801,306 People** |
| **Contribution Amount** | **TRY 351,406,489,608** |

Sources: Pension Monitoring Center, www.egm.org.tr, accessed on 19 March 2023.

## 4. Automatic Enrollment System

To increase the momentum, savings level, and fund size of the PPS, an automatic enrollment system was introduced in 2017 with Law No. 6740, taking examples from global practices. The automatic enrollment system is designed to include individuals in the system compulsorily when they first start working but allows flexibility in opting out. The

system is structured to gradually include participants based on the number of employees in the private and public sectors. The system targets employees under the age of 45. The retirement conditions are the same as those in the PPS. Furthermore, for those who do not exercise their right to opt out and receive the 30% government contribution, an additional initial contribution amount of TRY 1000 is offered. Similarly, individuals who choose to receive their accumulated savings as an annual income insurance upon retirement are eligible for an additional 5% government contribution. The entitlement periods are the same as in the voluntary PPS.

The design of the automatic enrollment system assumes that individuals will continue to stay in the system once they are enrolled (Kaya 2016). Italy stands out as the most successful example of automatic enrollment systems worldwide. Moreover, countries such as the United Kingdom, the United States, and Australia and Latin American countries like Chile have also implemented automatic enrollment systems for individual retirement (Kaya 2016).

However, despite successful pilot studies conducted in various cities and businesses prior to the system's launch, the opt-out rate from the automatic enrollment system nationwide, including both the public and private sectors, reached a significantly high level of approximately 70%. Although the reasons vary, generally low financial literacy among individuals in the system, a lack of perception of the system as a necessity, negative past experiences, low income levels of individuals, the presence of existing pension contracts, low fund returns, the absence of employer contributions, the low and noncumulative nature of government contributions, and problems in perception management have played significant roles in the high opt-out rates (Umut 2020).

Therefore, to increase the momentum and attractiveness of the individual pension system in the following years, two important reforms have been implemented. The first one is the regulation introduced on 25 May 2021 allowing parents to make pension contributions on behalf of minors under the age of 18 (Law on Amendment of the Insurance Law and Some Other Laws and a Decree-Law 2021). The second important reform is the increase in the government contribution rate to 30% on 22 January 2022, as mentioned above (Law on Amendment of the Law on Private Pension Savings and Investment System 2022).

As can be seen, recent reforms have been implemented in the individual pension system, and innovations have been introduced to increase interest in the system. The underlying aim is to increase the fund size and the number of participants in the PPS, thereby enhancing long-term savings levels and investments in the country.

## 5. Literature Review

In the conducted literature review, it has been observed that the data envelopment analysis (DEA) method is predominantly used to measure the efficiency of individual pension companies. While some studies categorized private pension system (PPS) companies separately, in certain studies (postgraduate theses), they were examined together with life insurance companies, and one study measured the effectiveness of the automatic enrollment system. However, no study was found that used the gray relational analysis method to measure the efficiency of individual pension companies.

According to the study conducted by Karakaya et al. (2014), the efficiency of individual pension companies was analyzed using the DEA method based on 2011 data, and it was determined that only three companies emerged as efficient, while others were found to operate suboptimally.

Sezgin and Yıldırım (2015) conducted a study to evaluate the efficiency of the PPS based on key indicators such as the number of individual pension contracts, fund amounts, and participation figures. However, no statistical analysis was performed in this study.

Ova (2018) analyzed the efficiency of individual pension companies using the DEA method for 3 years before and after the government contribution scheme. The results revealed that the companies did not effectively utilize sector resources and that the implementation of the government contribution negatively affected sector efficiency.

Göktolga and Karakış (2018) measured the efficiency of PPS companies using the AHP and VIKOR analyses based on data from 2014–2016. It was found that certain companies were efficient, but their financial performance fluctuated over time.

A study conducted by Acer et al. (2020) analyzed the efficiency of individual pension companies based on contribution amounts, fund sizes, participant numbers, and technical expense variables using the entropy and COPRAS methods. The results showed that the most significant variable for participants was the fund size, and three companies were found to be efficient in their operations.

Another study using the DEA method (Küçükkıralı and Aydın 2022) found that the efficiency of individual pension companies during 2014–2019 was 64%, indicating that they were not sufficiently efficient. However, fund management was calculated to be 94% efficient.

Moreover, in postgraduate thesis studies, it was observed that analyses using the DEA method were conducted to measure the efficiency of individual pension, automatic enrollment system, and life insurance companies at specific dates.

Only in the study conducted by Elitaş et al. (2012) was the financial performance of seven insurance companies traded on the stock exchange measured using the gray relational analysis method. However, this study was not specifically focused on individual pension companies; it was carried out on seven life and non-life insurance companies listed on the stock exchange, with only one individual pension company included.

In summary, various analyses are used to measure the efficiency or performance of individual pension companies, with the DEA method being predominantly employed. However, no study was found that used the gray relational analysis method, which is the focus of the present study, to measure the efficiency or performance of all pension companies in Türkiye.

Moreover, the DEA has certain limitations in performance measurement. It measures relative efficiency based on a specific set of observations. The inclusion of excessively large or small input–output values in the set of observations can create issues with determining the efficiency frontier. Additionally, DEA tends to identify the maximum point, so even a small deviation can lead the analyst astray from the results (Karasoy 2020). In contrast, the gray relational analysis method does not possess the mentioned weaknesses. Furthermore, gray relational analysis is not only used for performance measurement in the financial sector but also in various other fields such as aviation, transportation, and supply chain, providing innovative and effective results (Ayaydin and Durmuş 2015).

## 6. Method

### 6.1. Research Model

In recent years, the individual pension system, which has become increasingly important for the public, employers, and individuals, has emerged as a significant product of today's society. As the popularity of this system continues to grow, there is a need for studies that assess the performance, trends, and importance of individual pension systems for companies. This research aims to analyze the performance of insurance companies operating the individual pension system in Türkiye. The study will reveal the changes in the performance of insurance companies involved in the production of the individual pension system in Türkiye over the years.

### 6.2. Population and Sample

The research population consists of the 65 insurance companies operating in Türkiye as of 2022. As a sample, insurance companies that have engaged in the production of the individual pension system between the years 2016 and 2022, specifically during the months of December, have been selected.

### 6.3. Data Collection Tool

The selected sample consists of 15 insurance companies that have demonstrated continuity in their data in the period between 2016 and 2022, shown in the Table 4. The data pertaining to these companies were obtained from the statistical tables available on the official websites of the Insurance Association of Türkiye and the Pension Monitoring Center.

**Table 4.** Insurance Companies Included in the Research.

| No. | Company Title | Company Code |
|---|---|---|
| 1 | Aegon Emeklilik ve Hayat AŞ | A1 |
| 2 | Allianz Hayat ve Emeklilik AŞ | A2 |
| 3 | Allianz Yaşam ve Emeklilik AŞ | A3 |
| 4 | Anadolu Hayat Emeklilik AŞ | A4 |
| 5 | Bereket Emeklilik ve Hayat AŞ | A5 |
| 6 | AgeSA Emeklilik ve Hayat AŞ | A6 |
| 7 | Axa Hayat ve Emeklilik AŞ | A7 |
| 8 | BNP Paribas Cardif Emeklilik AŞ | A8 |
| 9 | Cigna Finans Emeklilik ve Hayat AŞ | A9 |
| 10 | Fiba Emeklilik ve Hayat AŞ | A10 |
| 11 | Garanti Emeklilik ve Hayat AŞ | A11 |
| 12 | Katılım Emeklilik ve Hayat AŞ | A12 |
| 13 | Metlife Emeklilik ve Hayat AŞ | A13 |
| 14 | NN Hayat ve Emeklilik AŞ | A14 |
| 15 | Türkiye Hayat ve Emeklilik AŞ * | A15 |

* Türkiye Hayat ve Emeklilik A.Ş., due to a company merger that occurred in 2020, the data from 2016 to 2019 were combined with the data of the other three companies and included in the dataset. Groupama Emeklilik A.Ş. was not included in the dataset because it ceased its individual pension activities by transferring its investment funds to Fiba Emeklilik ve Hayat A.Ş. in 2018.

A total of six variables were used in the study. These variables are shown in Table 5.

**Table 5.** Variables Used in the Research.

| Variable Name | Variable Code |
|---|---|
| Equity | D1 |
| Total Assets | D2 |
| Number of Participants | D3 |
| Total Participant Fund Amount | D4 |
| Pension Technical Revenue | D5 |
| Pension Technical Profit/Loss | D6 |

Since pension companies are a significant part of financial sector institutions, when evaluating companies' financial conditions, their financial capacities and aspects of financial sustainability, particularly the quality of equity and asset structure within the scope of sustainability, should be examined. The stronger a company's equity, the stronger its level of financial soundness. Similarly, the size of the assets represents all of the company's possessions and indicates its financial strength, borrowing capacity, debt repayment ability, and potential for new investments, thus reflecting the company's balance sheet size. For pension companies, the number of participants is crucial in terms of both new fund size and market share. A higher number of participants indicates that the company is preferred by customers and is more innovative in terms of product design, giving it an advantage over other companies. The total participant fund amount is tied to the contributions made by participants; the higher this amount, the more the company channels into investments. Technical revenue illustrates how much income pension companies derive from their insurance activities; a higher ratio signifies obtaining more participants and fund amounts while also implying fewer withdrawals from the system. Lastly, technical profit/loss indicates the profit or loss derived from the insurance activities of a pension company.

As seen, these ratios are highly useful criteria for measuring the performance of a pension company. Despite the numerous indicators in the insurance sector, these indicators are considered sufficient for performance measurement.

*6.4. Data Analysis*

In the research, gray relational analysis (GRA), which is one of the multi-criteria decision-making techniques, was employed. Gray relational analysis (GRA), a subcategory of gray modeling, was initially developed by Deng Ju-Long (Kula et al. 2016, p. 42). This analysis method converts uncertain situations into a simple and understandable numerical model (Aydemir et al. 2013, p. 188). Gray relational analysis consists of six steps (Demir et al. 2020, pp. 161–63):

**Step 1:** Construction of the decision matrix

$$X = \begin{vmatrix} x_{11}x_{12}x_{1n} \\ x_{21}x_{22}x_{2n} \\ x_{m1}x_{m2}x_{mn} \end{vmatrix}$$

**Step 2:** Determination of the reference series and comparison matrix

The reference series $x_0 = (x_0(j))$ is defined as follows for $j = 1, 2, \ldots, n$.

$x_0(j)$; $j$. The highest value in the normalized matrix is added as the reference series in the first row of the decision matrix, thus creating the comparison matrix.

**Step 3:** Normalization of the decision matrix

If the values in the matrix have a positive contribution to reaching the desired level when they are larger, the following equation is obtained:

$$x_i^* = \frac{x_i(j) - min\ x_i(j)}{max\ x_i(j) - min\ x_i(j)}$$

After these operations, the decision matrix is normalized, and a new matrix X* is formed.

**Step 4:** Calculation of the absolute value matrix

A new matrix is formed by taking the absolute value of the difference between $x_0$ and $x_i^*$.

**Step 5:** Determination of the gray relational coefficient matrix

$$Y_{0i}(j) = \frac{\Delta_{min} + \zeta\Delta_{max}}{\Delta_{0i}(j) + \zeta\Delta_{max}}$$

where $\Delta_{0i}(j)$: $\Delta_i$ represents the $j$ value in the difference data sequence. $\zeta$: is discrimination coefficient, ranging from 0 to 1. It is commonly taken as 0.5.

**Step 6:** Calculation of the gray relational degrees

They are calculated as $r_{0i} = \frac{1}{n}\sum_{j=1}^{n} Y_{0i}(j)$ and $i = 1, 2, 3\ldots, m$.

In this section of the study, the performance of the mentioned insurance companies in the individual retirement systems between 2016 and 2022 has been analyzed. In this regard, the following findings have been shared. As an example, only the gray relational analysis solution for the second quarter of 2022 will be demonstrated in the study, and the ranking over the years will be provided at the end.

**Step 1:** Formation of the Decision Matrix (X). The decision matrix is created in Table 6.

**Table 6.** Decision Matrix.

| COMPANIES | D1 | D2 | D3 | D4 | D5 | D6 |
|---|---|---|---|---|---|---|
| A1 | 436,597,485 | 8,922,230,670 | 34,744 | 160,937,001 | 1,262,475 | −9,208,962 |
| A2 | 113,925,545 | 9,938,228,257 | 672,363 | 45,636,934,225 | 67,715,684 | −163,094 |
| A3 | 1,639,777,716 | 45,244,719,758 | 88,196 | 8,051,549,682 | 337,184,826 | 109,517,411 |
| A4 | 2,424,616,527 | 62,985,771,631 | 792,250 | 35,570,370,999 | 453,585,222 | 84,976,318 |
| A5 | 181,823,413 | 1,776,759,873 | 1,211,274 | 44,443,570,448 | 7,979,158 | −1,848,158 |
| A6 | 1,195,928,727 | 61,608,212,238 | 35,192 | 1,242,015,513 | 408,036,825 | −83,088,984 |
| A7 | 47,910,302 | 1,810,832,449 | 84,613 | 1,000,087,805 | 11,923,974 | −10,081,777 |
| A8 | 694,699,575 | 8,166,103,379 | 184,101 | 5,821,729,054 | 63,559,707 | −6,364,512 |
| A9 | 216,977,178 | 3,067,434,421 | 59,748 | 1,274,671,091 | 8,985,587 | −1,849,242 |
| A10 | 123,381,602 | 5,615,912,858 | 94,548 | 4,355,389,984 | 35,596,963 | 4,605,370 |
| A11 | 1,435,160,926 | 45,232,092,318 | 1,194,998 | 34,509,040,602 | 339,230,582 | 98,499,064 |
| A12 | 239,438,880 | 8,005,696,891 | 372,957 | 5,964,849,125 | 82,761,927 | 28,292,126 |
| A13 | 1,007,273,242 | 7,121,004,210 | 200,886 | 3,481,356,105 | 55,070,272 | 19,856,240 |
| A14 | 149,878,844 | 10,662,741,489 | 254,043 | 8,646,036,369 | 88,438,366 | 12,104,952 |
| A15 | 3,957,106,196 | 77,000,196,262 | 2,174,259 | 47,669,407,344 | 530,862,848 | 222,714,607 |

**Step 2:** Determination of the reference series and comparison matrix (shown in the Table 7).

**Table 7.** Decision Matrix with Added Reference Series.

| COMPANIES | D1 | D2 | D3 | D4 | D5 | D6 |
|---|---|---|---|---|---|---|
| Reference Series | 1000 | 1000 | 1000 | 1000 | 1000 | 1000 |
| A1 | 436,597,485 | 8,922,230,670 | 34,744 | 160,937,001 | 1,262,475 | −9,208,962 |
| A2 | 113,925,545 | 9,938,228,257 | 672,363 | 45,636,934,225 | 67,715,684 | −163,094 |
| A3 | 1,639,777,716 | 45,244,719,758 | 88,196 | 8,051,549,682 | 337,184,826 | 109,517,411 |
| A4 | 2,424,616,527 | 62,985,771,631 | 792,250 | 35,570,370,999 | 453,585,222 | 84,976,318 |
| A5 | 181,823,413 | 1,776,759,873 | 1,211,274 | 44,443,570,448 | 7,979,158 | −1,848,158 |
| A6 | 1,195,928,727 | 61,608,212,238 | 35,192 | 1,242,015,513 | 408,036,825 | −83,088,984 |
| A7 | 47,910,302 | 1,810,832,449 | 84,613 | 1,000,087,805 | 11,923,974 | −10,081,777 |
| A8 | 694,699,575 | 8,166,103,379 | 184,101 | 5,821,729,054 | 63,559,707 | −6,364,512 |
| A9 | 216,977,178 | 3,067,434,421 | 59,748 | 1,274,671,091 | 8,985,587 | −1,849,242 |
| A10 | 123,381,602 | 5,615,912,858 | 94,548 | 4,355,389,984 | 35,596,963 | 4,605,370 |
| A11 | 1,435,160,926 | 45,232,092,318 | 1,194,998 | 34,509,040,602 | 339,230,582 | 98,499,064 |
| A12 | 239,438,880 | 8,005,696,891 | 372,957 | 5,964,849,125 | 82,761,927 | 28,292,126 |
| A13 | 1,007,273,242 | 7,121,004,210 | 200,886 | 3,481,356,105 | 55,070,272 | 19,856,240 |
| A14 | 149,878,844 | 10,662,741,489 | 254,043 | 8,646,036,369 | 88,438,366 | 12,104,952 |
| A15 | 3,957,106,196 | 77,000,196,262 | 2,174,259 | 47,669,407,344 | 530,862,848 | 222,714,607 |

The values of the reference series are taken based on the highest value in that column. The determined reference series is as follows:

$X_0$ = {3,957,106,196/77,000,196,262/2,174,259/47,669,407,344/530,862,848/222,714,607}

**Step 3:** Normalization of the Decision Matrix (shown in the Table 8).

X1*(1) = (436,597,485 − 47,910,302)/(3,957,106,196 − 47,910,302) = 0.099

**Step 4:** Creation of the Absolute Value Matrix (shown in the Table 9).
The difference between the reference series and the ordinal values is calculated.

X1 = (1.000 − 0.406) = 0.594

**Step 5:** Determination of Gray Relational Coefficient Matrix (shown in the Table 10).

$$\gamma_{0i}(j) = \frac{0 + 1(0.5)}{0.901 + 1(0.5)} = 0.357$$

**Step 6:** Calculation of Gray Relational Degrees (shown in the Table 11).

The arithmetic mean of each row is calculated.

$$r_{01} = \frac{(0.357 + 0.356 + 0.333 + 0.333 + 0.333 + 0.397)}{6} = 0.352$$

**Table 8.** Normalized Decision Matrix.

| COMPANIES | D1 | D2 | D3 | D4 | D5 | D6 |
|---|---|---|---|---|---|---|
| A1 | 0.099 | 0.095 | 0.000 | 0.000 | 0.000 | 0.242 |
| A2 | 0.017 | 0.108 | 0.298 | 0.957 | 0.125 | 0.271 |
| A3 | 0.407 | 0.578 | 0.025 | 0.166 | 0.634 | 0.630 |
| A4 | 0.608 | 0.814 | 0.354 | 0.745 | 0.854 | 0.550 |
| A5 | 0.034 | 0.000 | 0.550 | 0.932 | 0.013 | 0.266 |
| A6 | 0.294 | 0.795 | 0.000 | 0.023 | 0.768 | 0.000 |
| A7 | 0.000 | 0.000 | 0.023 | 0.018 | 0.020 | 0.239 |
| A8 | 0.165 | 0.085 | 0.070 | 0.119 | 0.118 | 0.251 |
| A9 | 0.043 | 0.017 | 0.012 | 0.023 | 0.015 | 0.266 |
| A10 | 0.019 | 0.051 | 0.028 | 0.088 | 0.065 | 0.287 |
| A11 | 0.355 | 0.578 | 0.542 | 0.723 | 0.638 | 0.594 |
| A12 | 0.049 | 0.083 | 0.158 | 0.122 | 0.154 | 0.364 |
| A13 | 0.245 | 0.071 | 0.078 | 0.070 | 0.102 | 0.337 |
| A14 | 0.026 | 0.118 | 0.102 | 0.179 | 0.165 | 0.311 |
| A15 | 1.000 | 1.000 | 1.000 | 1.000 | 1.000 | 1.000 |

**Table 9.** Absolute Value Matrix.

| COMPANIES | D1 | D2 | D3 | D4 | D5 | D6 |
|---|---|---|---|---|---|---|
| A1 | 0.901 | 0.905 | 1.000 | 1.000 | 1.000 | 0.758 |
| A2 | 0.983 | 0.892 | 0.702 | 0.043 | 0.875 | 0.729 |
| A3 | 0.593 | 0.422 | 0.975 | 0.834 | 0.366 | 0.370 |
| A4 | 0.392 | 0.186 | 0.646 | 0.255 | 0.146 | 0.450 |
| A5 | 0.966 | 1.000 | 0.450 | 0.068 | 0.987 | 0.734 |
| A6 | 0.706 | 0.205 | 1.000 | 0.977 | 0.232 | 1.000 |
| A7 | 1.000 | 1.000 | 0.977 | 0.982 | 0.980 | 0.761 |
| A8 | 0.835 | 0.915 | 0.930 | 0.881 | 0.882 | 0.749 |
| A9 | 0.957 | 0.983 | 0.988 | 0.977 | 0.985 | 0.734 |
| A10 | 0.981 | 0.949 | 0.972 | 0.912 | 0.935 | 0.713 |
| A11 | 0.645 | 0.422 | 0.458 | 0.277 | 0.362 | 0.406 |
| A12 | 0.951 | 0.917 | 0.842 | 0.878 | 0.846 | 0.636 |
| A13 | 0.755 | 0.929 | 0.922 | 0.930 | 0.898 | 0.663 |
| A14 | 0.974 | 0.882 | 0.898 | 0.821 | 0.835 | 0.689 |
| A15 | 0.000 | 0.000 | 0.000 | 0.000 | 0.000 | 0.000 |

**Table 10.** Gray Relational Coefficient Matrix.

| COMPANIES | D1 | D2 | D3 | D4 | D5 | D6 |
|---|---|---|---|---|---|---|
| A1 | 0.357 | 0.356 | 0.333 | 0.333 | 0.333 | 0.397 |
| A2 | 0.337 | 0.359 | 0.416 | 0.921 | 0.364 | 0.407 |
| A3 | 0.458 | 0.542 | 0.339 | 0.375 | 0.578 | 0.575 |
| A4 | 0.561 | 0.729 | 0.436 | 0.663 | 0.774 | 0.526 |
| A5 | 0.341 | 0.333 | 0.526 | 0.880 | 0.336 | 0.405 |
| A6 | 0.414 | 0.710 | 0.333 | 0.338 | 0.683 | 0.333 |
| A7 | 0.333 | 0.333 | 0.339 | 0.337 | 0.338 | 0.396 |
| A8 | 0.375 | 0.353 | 0.350 | 0.362 | 0.362 | 0.400 |
| A9 | 0.343 | 0.337 | 0.336 | 0.339 | 0.337 | 0.405 |
| A10 | 0.338 | 0.345 | 0.340 | 0.354 | 0.348 | 0.412 |
| A11 | 0.437 | 0.542 | 0.522 | 0.643 | 0.580 | 0.552 |
| A12 | 0.345 | 0.353 | 0.373 | 0.363 | 0.371 | 0.440 |
| A13 | 0.399 | 0.350 | 0.352 | 0.350 | 0.358 | 0.430 |
| A14 | 0.339 | 0.362 | 0.358 | 0.378 | 0.374 | 0.421 |
| A15 | 1.000 | 1.000 | 1.000 | 1.000 | 1.000 | 1.000 |

**Table 11.** Gray Relational Degrees and Rankings.

| COMPANIES | Gray Relational Ranking | Ranking |
|---|---|---|
| **A1** | 0.352 | 13 |
| **A2** | 0.467 | 7 |
| **A3** | 0.478 | 4 |
| **A4** | 0.615 | 2 |
| **A5** | 0.470 | 5 |
| **A6** | 0.469 | 6 |
| **A7** | 0.346 | 15 |
| **A8** | 0.367 | 11 |
| **A9** | 0.349 | 14 |
| **A10** | 0.356 | 12 |
| **A11** | 0.546 | 3 |
| **A12** | 0.374 | 8 |
| **A13** | 0.373 | 9 |
| **A14** | 0.372 | 10 |
| **A15** | 1.000 | 1 |

## 7. Results

Table 12 presents the values obtained from the gray relational analysis results for the insurance companies evaluated in the study, categorized by years. For seven years, the top three companies with the highest values were Türkiye Hayat ve Emeklilik A.Ş., Garanti Emeklilik ve Hayat A.Ş., and Anadolu Hayat Emeklilik A.Ş. On the other hand, the three companies with the lowest values were Aegon Emeklilik ve Hayat A.Ş., Cigna Finans Emeklilik ve Hayat A.Ş., and Axa Hayat ve Emeklilik A.Ş.

**Table 12.** Rankings of Insurance Companies for the Years 2016–2022.

| Value | 2016(4) Ranking | Value | 2017(4) Ranking | Value | 2018(4) Ranking | Value | 2019(4) Ranking | Value | 2020(4) Ranking | Value | 2021(4) Ranking | Value | 2022(2) Ranking |
|---|---|---|---|---|---|---|---|---|---|---|---|---|---|
| 0.814 | **A11** | 0.864 | **A15** | 0.920 | **A15** | 0.896 | **A15** | 0.994 | **A15** | 1.000 | **A15** | 1.000 | **A15** |
| 0.776 | **A4** | 0.791 | **A11** | 0.735 | **A4** | 0.698 | **A4** | 0.683 | **A4** | 0.619 | **A4** | 0.615 | **A4** |
| 0.716 | **A15** | 0.723 | **A4** | 0.671 | **A11** | 0.651 | **A11** | 0.560 | **A11** | 0.552 | **A11** | 0.546 | **A11** |
| 0.685 | **A6** | 0.552 | **A6** | 0.534 | **A6** | 0.523 | **A3** | 0.526 | **A3** | 0.477 | **A2** | 0.478 | **A3** |
| 0.527 | **A3** | 0.489 | **A3** | 0.524 | **A3** | 0.492 | **A6** | 0.466 | **A5** | 0.476 | **A5** | 0.470 | **A5** |
| 0.390 | **A13** | 0.484 | **A5** | 0.476 | **A5** | 0.471 | **A5** | 0.463 | **A6** | 0.462 | **A6** | 0.469 | **A6** |
| 0.367 | **A14** | 0.385 | **A13** | 0.379 | **A13** | 0.380 | **A13** | 0.373 | **A13** | 0.462 | **A3** | 0.467 | **A2** |
| 0.367 | **A5** | 0.366 | **A14** | 0.361 | **A14** | 0.363 | **A12** | 0.362 | **A12** | 0.374 | **A13** | 0.374 | **A12** |
| 0.367 | **A2** | 0.365 | **A2** | 0.361 | **A2** | 0.360 | **A8** | 0.361 | **A8** | 0.371 | **A12** | 0.373 | **A13** |
| 0.364 | **A8** | 0.360 | **A8** | 0.358 | **A8** | 0.359 | **A14** | 0.359 | **A14** | 0.369 | **A14** | 0.372 | **A14** |
| 0.355 | **A9** | 0.352 | **A12** | 0.357 | **A12** | 0.357 | **A2** | 0.354 | **A2** | 0.363 | **A8** | 0.367 | **A8** |
| 0.351 | **A12** | 0.352 | **A9** | 0.351 | **A10** | 0.353 | **A10** | 0.350 | **A10** | 0.356 | **A10** | 0.356 | **A10** |
| 0.346 | **A10** | 0.345 | **A10** | 0.350 | **A9** | 0.348 | **A9** | 0.344 | **A9** | 0.350 | **A9** | 0.352 | **A1** |
| 0.344 | **A7** | 0.343 | **A7** | 0.342 | **A7** | 0.342 | **A7** | 0.340 | **A1** | 0.349 | **A1** | 0.349 | **A9** |
| 0.342 | **A1** | 0.340 | **A1** | 0.339 | **A1** | 0.340 | **A1** | 0.339 | **A7** | 0.346 | **A7** | 0.346 | **A7** |

To provide a clearer view of the changes in the performance rankings of companies relative to each other over the years, Table 13 was created.

The table presents the ranking based on the values obtained from the analysis of insurance companies. As evident from this ranking, Türkiye Hayat ve Emeklilik A.Ş. has achieved the highest performance between the years 2016 and 2022, while Axa Hayat ve Emeklilik A.Ş. has the lowest performance.

**Table 13.** Rankings of Insurance Companies for the Years 2016–2022.

| COMPANIES | 2016(4) | 2017(4) | 2018(4) | 2019(4) | 2020(4) | 2021(4) | 2022(2) |
|---|---|---|---|---|---|---|---|
| A1 | 15 | 15 | 15 | 15 | 14 | 14 | 13 |
| A2 | 9 | 9 | 9 | 11 | 11 | 4 | 7 |
| A3 | 5 | 5 | 5 | 4 | 4 | 7 | 4 |
| A4 | 2 | 3 | 2 | 2 | 2 | 2 | 2 |
| A5 | 8 | 6 | 6 | 6 | 5 | 5 | 5 |
| A6 | 4 | 4 | 4 | 5 | 6 | 6 | 6 |
| A7 | 14 | 14 | 14 | 14 | 15 | 15 | 15 |
| A8 | 10 | 10 | 10 | 9 | 9 | 11 | 11 |
| A9 | 11 | 12 | 13 | 13 | 13 | 13 | 14 |
| A10 | 13 | 13 | 12 | 12 | 12 | 12 | 12 |
| A11 | 1 | 2 | 3 | 3 | 3 | 3 | 3 |
| A12 | 12 | 11 | 11 | 8 | 8 | 9 | 8 |
| A13 | 6 | 7 | 7 | 7 | 7 | 8 | 9 |
| A14 | 7 | 8 | 8 | 10 | 10 | 10 | 10 |
| 15 | 3 | 1 | 1 | 1 | 1 | 1 | 1 |

## 8. Comparison of Results with Data Envelopment Analysis

Although the gray relational analysis method produces stronger and more reliable results compared to the DEA, it is deemed beneficial to compare it with DEA, widely used for performance measurement in practice.

After applying the output-oriented CCR model of data envelopment analysis, the efficiency scores and findings obtained were presented, followed by the presentation of the findings obtained through the gray relational analysis method. In data envelopment analysis, positive correlations between input and output variables are expected. It is believed that this positive relationship will enhance the reliability of the analysis (Akkurt and Okur 2022). Values ranging from 0.71 to 0.99 indicate a high level of correlation. For this purpose, the inputs and outputs of the companies operating in the individual pension system were subjected to Pearson correlation analysis using Excel.

According to the gray relational analysis method, the analysis results for the performance measurement between 2016 and 2021 are provided in Table 14.

**Table 14.** Rankings of Insurance Companies for the Years 2016–2022.

| | 2016 | | 2017 | | 2018 | | 2019 | | 2020 | | 2021 | | 2022/2 |
|---|---|---|---|---|---|---|---|---|---|---|---|---|---|
| 1 | 0.814 | A11 | 0.864 | A15 | 0.920 | A15 | 0.896 | A15 | 0.994 | A15 | 1.000 | A15 | 1.000 | A15 |
| 2 | 0.776 | A4 | 0.791 | A11 | 0.735 | A4 | 0.698 | A4 | 0.683 | A4 | 0.619 | A4 | 0.615 | A4 |
| 3 | 0.716 | A15 | 0.723 | A4 | 0.671 | A11 | 0.651 | A11 | 0.560 | A11 | 0.552 | A11 | 0.546 | A11 |
| 4 | 0.685 | A6 | 0.552 | A6 | 0.534 | A6 | 0.523 | A6 | 0.526 | A3 | 0.477 | A2 | 0.478 | A3 |
| 5 | 0.527 | A3 | 0.489 | A3 | 0.524 | A3 | 0.492 | A3 | 0.466 | A5 | 0.476 | A5 | 0.470 | A5 |
| 6 | 0.390 | A13 | 0.484 | A5 | 0.476 | A5 | 0.471 | A5 | 0.463 | A6 | 0.462 | A3 | 0.469 | A6 |
| 7 | 0.367 | A2 | 0.365 | A2 | 0.379 | A13 | 0.380 | A13 | 0.373 | A13 | 0.462 | A6 | 0.467 | A2 |
| 8 | 0.367 | A5 | 0.366 | A14 | 0.361 | A2 | 0.363 | A12 | 0.362 | A12 | 0.374 | A13 | 0.374 | A12 |
| 9 | 0.367 | A14 | 0.385 | A13 | 0.361 | A14 | 0.360 | A8 | 0.361 | A8 | 0.371 | A12 | 0.373 | A13 |
| 10 | 0.364 | A8 | 0.360 | A8 | 0.358 | A8 | 0.359 | A14 | 0.359 | A14 | 0.369 | A14 | 0.372 | A14 |
| 11 | 0.355 | A9 | 0.352 | A12 | 0.357 | A12 | 0.357 | A2 | 0.354 | A2 | 0.363 | A8 | 0.367 | A8 |
| 12 | 0.351 | A12 | 0.352 | A9 | 0.351 | A10 | 0.353 | A10 | 0.350 | A10 | 0.356 | A10 | 0.356 | A10 |
| 13 | 0.346 | A10 | 0.345 | A10 | 0.350 | A9 | 0.348 | A9 | 0.344 | A9 | 0.350 | A9 | 0.352 | A1 |
| 14 | 0.344 | A7 | 0.343 | A7 | 0.342 | A7 | 0.342 | A7 | 0.340 | A1 | 0.349 | A1 | 0.349 | A9 |
| 15 | 0.342 | A1 | 0.340 | A1 | 0.339 | A1 | 0.340 | A1 | 0.339 | A7 | 0.346 | A7 | 0.346 | A7 |

Table 14 provides the performance measurement values of the insurance companies evaluated in the study using the gray relational method according to years. Throughout the 6-year period, the three companies with the highest values each year were Turkey Life and Pension Inc., Garanti Pension and Life Inc., and Anadolu Life and Pension Inc. On the other hand, the company with the lowest value between 2016 and 2019 was Aegon Pension and Life Inc., while in 2020 and 2021, it was Axa Life and Pension Inc.

Table 15 presents the rankings of the companies that are efficient according to the DEA method in terms of their performance based on the gray relational analysis method.

**Table 15.** Rankings of DEA and Gray Relational Analysis Method Results.

| Company Title | 2016 DEA | 2016 GRAY RELATIONAL ANALYSIS |
|---|---|---|
| Garanti Emeklilik ve Hayat A.Ş. | 1 | 0.814 |
| Anadolu Hayat Emeklilik A.Ş. | 1 | 0.776 |
| Türkiye Hayat ve Emeklilik A.Ş. | 1 | 0.716 |
| Metlife Emeklilik ve Hayat A.Ş. | 1 | 0.390 |
| Fiba Emeklilik ve Hayat A.Ş. | 1 | 0.346 |
| Axa Hayat ve Emeklilik A.Ş. | 1 | 0.344 |
| Aegon Emeklilik ve Hayat A.Ş. | 1 | 0.342 |
| | **2017 DEA** | **2017 GRAY RELATIONAL ANALYSIS** |
| Türkiye Hayat ve Emeklilik A.Ş. | 1 | 0.864 |
| Garanti Emeklilik ve Hayat A.Ş. | 1 | 0.791 |
| Anadolu Hayat Emeklilik A.Ş. | 1 | 0.723 |
| AgeSA Emeklilik ve Hayat A.Ş. | 1 | 0.552 |
| Fiba Emeklilik ve Hayat A.Ş. | 1 | 0.345 |
| Axa Hayat ve Emeklilik A.Ş. | 1 | 0.343 |
| Aegon Emeklilik ve Hayat A.Ş. | 1 | 0.340 |
| | **2018 DEA** | **2018 GRAY RELATIONAL ANALYSIS** |
| Türkiye Hayat ve Emeklilik A.Ş. | 1 | 0.920 |
| Anadolu Hayat Emeklilik A.Ş. | 1 | 0.735 |
| Garanti Emeklilik ve Hayat A.Ş. | 1 | 0.671 |
| AgeSA Emeklilik ve Hayat A.Ş. | 1 | 0.534 |
| Axa Hayat ve Emeklilik A.Ş. | 1 | 0.342 |
| Aegon Emeklilik ve Hayat A.Ş. | 1 | 0.339 |
| | **2019 DEA** | **2019 GRAY RELATIONAL ANALYSIS** |
| Türkiye Hayat ve Emeklilik A.Ş. | 1 | 0.896 |
| Anadolu Hayat Emeklilik A.Ş. | 1 | 0.698 |
| Garanti Emeklilik ve Hayat A.Ş. | 1 | 0.651 |
| AgeSA Emeklilik ve Hayat A.Ş. | 1 | 0.492 |
| Axa Hayat ve Emeklilik A.Ş. | 1 | 0.342 |
| Aegon Emeklilik ve Hayat A.Ş. | 1 | 0.340 |
| | **2020 DEA** | **2020 GRAY RELATIONAL ANALYSIS** |
| Türkiye Hayat ve Emeklilik A.Ş. | 1 | 0.994 |
| Anadolu Hayat Emeklilik A.Ş. | 1 | 0.683 |
| Garanti Emeklilik ve Hayat A.Ş. | 1 | 0.560 |
| AgeSA Emeklilik ve Hayat A.Ş. | 1 | 0.463 |
| Cigna Finans Emeklilik ve Hayat A.Ş. | 1 | 0.344 |
| Aegon Emeklilik ve Hayat A.Ş. | 1 | 0.340 |
| Axa Hayat ve Emeklilik A.Ş. | 1 | 0.339 |

**Table 15.** *Cont.*

|  | 2021 DEA | 2021 GRAY RELATIONAL ANALYSIS |
|---|---|---|
| **Türkiye Hayat ve Emeklilik A.Ş.** | 1 | 1.000 |
| **Garanti Emeklilik ve Hayat A.Ş.** | 1 | 0.552 |
| **Bereket Emeklilik ve Hayat A.Ş.** | 1 | 0.476 |
| **AgeSA Emeklilik ve Hayat A.Ş.** | 1 | 0.462 |
| **Fiba Emeklilik ve Hayat A.Ş.** | 1 | 0.356 |
| **Cigna Finans Emeklilik ve Hayat A.Ş.** | 1 | 0.350 |
| **Aegon Emeklilik ve Hayat A.Ş.** | 1 | 0.349 |
| **Axa Hayat ve Emeklilik A.Ş.** | 1 | 0.346 |
|  | **2022/2 DEA** | **2022/2 GRAY RELATIONAL ANALYSIS** |
| **Türkiye Hayat ve Emeklilik A.Ş** | 1 | 1.000 |
| **Garanti Emeklilik ve Hayat A.Ş.** | 1 | 0.546 |
| **AgeSA Emeklilik ve Hayat A.Ş.** | 1 | 0.469 |
| **Aegon Emeklilik ve Hayat A.Ş** | 1 | 0.352 |
| **Cigna Finans Emeklilik ve Hayat A.Ş.** | 1 | 0.349 |

Table 15 presents a ranking based on the results of the gray relational analysis method for companies that are efficient according to the DEA method. According to the results, Turkey Life and Pension Inc. exhibited the highest performance, while Aegon Pension and Life Inc. and Axa Life and Pension Inc. showed the lowest performances. In other words, the outcomes obtained from the gray relational analysis method are consistent with the results obtained through the data envelopment analysis method.

## 9. Conclusions

The private pension system (PPS) is a savings and investment system supported by government contributions, which provides the opportunity to accumulate regular savings to maintain current living standards during retirement. The system requires a minimum of 10 years of participation and meeting the age requirement of 56 in order to qualify for retirement benefits in Türkiye. The development of this system is not only important for participants and companies but also for the country's economy. The funds in this system not only stimulate the financial sector by directing investments but also encourage savings for individuals. Therefore, it is necessary to measure the effectiveness and performance of active companies in this system and take measures to enhance their effectiveness and performance based on these measurements. This study aims to contribute to the literature in the future and provide individuals with information about the companies in this system.

In the study, the performance of 15 insurance companies involved in the activities of the private pension system was measured using the gray relational analysis method in Türkiye. The performances of the companies under evaluation were analyzed using six different variables. When examining the current research, it was observed that data envelopment analysis is commonly used for efficiency and performance measurement. However, in this study, gray relational analysis was applied differently from other studies. The reason for this choice is that gray relational analysis can yield effective results even with a small amount of data, unlike methods like data envelopment analysis that require more data. Considering that the sample size of this study consists of 15 insurance companies, the gray relational analysis method was chosen.

According to the results of the analysis, there has been a decrease in the performance ratios of companies over the years, and some insurance companies have experienced a decline in the number of participants. However, in terms of company performance rankings, Türkiye Hayat ve Emeklilik A.Ş., Anadolu Hayat ve Emeklilik A.Ş., and Garanti Emeklilik ve Hayat A.Ş. consistently remained the top three. Türkiye Hayat ve Emeklilik A.Ş. was identified as the best-performing company according to the analysis, primarily because of

its effective management of equity, total assets, number of participants, participant fund amount, retirement technical income, and retirement technical profit/loss.

The findings of this study are consistent with previous research, which provides support for the validity of the results. In the study, the same variables were tested using the data envelopment analysis method for the specified years, and a similar outcome was achieved. This situation implies that the results are corroborated by other studies.

Based on the conducted analyses and evaluations, the recent decision to increase the government contribution to 30% and the establishment of individual pension plans for individuals under the age of 18 are highly positive developments. However, this government contribution alone may not be sufficient for the system. Therefore, businesses may need to incentivize their employees to stay in the system for longer periods. The best example of this is the automatic enrollment system, which was introduced in 2017. It is believed that this system can move the system to more advanced levels by ensuring that employees stay for extended periods. Additionally, funds need to be invested in more suitable investment instruments and their alternatives.

It should be noted that changing the number of variables used in the analysis would result in different analysis outcomes. Therefore, it is possible to conduct analyses using different variables or various technical ratios for the performance analysis of individual pension companies in our country, and this study sheds light on such possibilities.

**Funding:** This research received no external funding.

**Data Availability Statement:** All data has been obtained from the publicly available website of the Insurance Association of Türkiye. Here, data regarding all insurance and individual pension companies is published by year, available: https://www.tsb.org.tr/en/insurance-data-and-financial-tables (accessed on 19 March 2023).

**Conflicts of Interest:** The author declares no conflict of interest.

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
