# Peer review of "Measuring the Performance of Private Pension Companies in Türkiye by Gray Relational Analysis Method"

_jrfm, doi:10.3390/jrfm16090396_

Round 1

Reviewer 1 Report

This study uses the grey relational analysis method to analyze and evaluate the performance of 15 private pension insurance companies in Türkiye, which can play a certain reference role in promoting the standardized development of Türkiye's private pension market. The paper selected 6 indicators to calculate and rank the performance of 15 private pension insurance companies. Overall, the topic selection of the paper has certain practical significance. The sample selection is comprehensive, the methods are appropriate, the indicator system is sound, and the calculation and analysis results are fair. The main issues that need improvement include:

  (1) Further explain that the use of grey correlation analysis method for performance measurement and evaluation of private pension insurance companies is innovative, and there may be better performance measurement methods? (2) Further explain the suitability of the selected 6 indicators for performance evaluation measurement. Should the indicator system be more comprehensive or have a better indicator system? (3) The writing standards of the paper should be more comprehensive, such as excessive description of the background in the abstract, while the truly important research objectives, samples, methods, indicators, results, and suggestions are rarely reflected. Based on comprehensive evaluation, the paper is too simple and requires significant modifications before submission for review.

The overall English language expression ability of this paper is good. The standardization of the abstract writing and the standardization of the tables need to be further improved. If many tables are too wide, the layout can be improved.

Author Response

I would like to express my gratitude for the valuable insights and contributions provided by Reviewer. The observations made are quite pertinent. In line with the comments of Reviewer, the rationale behind the utilization of grey relational analysis and its strengths have been elucidated through additions to the "literature review" section. Furthermore, corresponding amendments have been introduced in the introduction part. Additionally, a new section (Section 8) has been incorporated into the study to compare the consistency and reliability of the grey relational analysis results using a separate statistical method known as Data Envelopment Analysis (DEA).

In line with Comment 2, pertinent augmentations and explanations have been introduced under the heading "6.3. Data Collection Tool," relating to the variables employed in the study. The significance of these indicators has also been expounded upon.

Regarding Comment 3, the introduction section has been reinforced, accompanied by appropriate additions.

In light of these enhancements, new references have been integrated into the References, as part of the aforementioned efforts.

Reviewer 2 Report

In this paper, authors aimed to measure the performance of individual pension companies operating in Türkiye using the Gray Relational Analysis Method, which is an effective measurement method, for the years 2016-2022. This paper investigates an interesting problem, and the structure is relative good. However, minor revisions are needed before the acceptance.

1. Since this is a measurement paper, have the authors tried other method besides Gray Relational Analysis.

2. How different cities impact the modeling and analysis, authors can make is clearer.

3. Please add a comparison table to cover all measured cities or areas in this paper.

4. Please go through the paper carefully and double check whether the right template are used. Correct some typos and formatting issues (e.g., “The Private Pension System (IPS)” -> “The Private Pension System (PPS)”?).

5. Make the References more comprehensive, besides this work, some other promising scenarios can be covered in this work. If the above related work can be discussed, it can strongly improve the research significance. For the improvement, the following papers can be considered to make the references more comprehensive.

Jingyu Zhang, Siqi Zhong, Jin Wang, Xiaofeng Yu, Osama Alfarraj. A Storage Optimization Scheme for Blockchain Transaction Databases. Computer Systems Science and Engineering, 2021, 36(3):521-535.

Jin Wang, Wencheng Chen, Yongjun Ren, Osama Alfarraj, Lei Wang. Blockchain Based Data Storage Mechanism in Cyber Physical System. Journal of Internet Technology, 2020, 21(6): 1681-1689

None

Author Response

I extend my sincere appreciation for the valuable insights and contributions provided by Reviewer. The comments offered demonstrate a keen understanding of the research. With respect to comment 1, the rationale behind the selection of grey relational analysis methodology has been comprehensively explained and supplemented. Notably, although the option of employing Data Envelopment Analysis (DEA) for performance measurement was considered, the reasons for its exclusion have been elaborated upon. However, a new Section 8 has been integrated into the study to facilitate the comparison of grey relational analysis results, utilizing the DEA as an alternative statistical method.

Comments 2 and 3 pertain to the context of the study. In a country with 81 cities, pension companies operate nationwide without providing statistical breakdowns by individual city. Given the extensive number of cities and the study's objective of measuring overall performance for all of Türkiye, a detailed breakdown by city has been intentionally omitted.

Addressing comment 4, the necessary revisions have been incorporated. It is noteworthy that special care has been taken to align with the journal's writing and formatting guidelines, particularly concerning tables.

Comment 5 has prompted the inclusion of new references in the manuscript.

Reviewer 3 Report

The study assessed the performance of 15 insurance companies participating in the activities of the private pension system using the well-known algorithm of the gray relational analysis method, taking into account the small sample. It was expedient to compare the obtained results with the application of some known methods of multi-criteria research.

As presented, the article is descriptive in nature, without substantiation of the reliability of the obtained results, without elements of novelty, and can be recommended for publication only with substantial revision.

Author Response

I‘d like to express my gratitude for the valuable insights and contributions provided by Reviewer. The recommendations put forth have been instrumental in enhancing the study. In response to the Reviewer's suggestion, the data has been subjected to analysis using the widely recognized Data Envelopment Analysis (DEA) method, in addition to the grey relational analysis, to ensure the robustness of the study's findings. The outcomes of the DEA analysis have exhibited consistency with the results obtained from the grey relational analysis.

Furthermore, in accordance with the Reviewer's advice, a new Section 8 has been seamlessly integrated into the study.

Once again, I extend my appreciation to all Reviewers for their valuable comments and contributions, which have undoubtedly enriched the quality of this study.

Round 2

Reviewer 3 Report

With the given edits and explanations, the article is recommended for publication